# Nutritional Value of Climate-Resilient Forage Species Sustaining Peri-Urban Dairy Cow Production in the Coastal Grasslands of Benin (West Africa)

**DOI:** 10.3390/ani12243550

**Published:** 2022-12-15

**Authors:** Bossima Ivan Koura, Alessandro Vastolo, Dieu donné Kiatti, Monica Isabella Cutrignelli, Marcel Houinato, Serena Calabrò

**Affiliations:** 1Ecole de Gestion et d’Exploitation de Système d’Elevage, Université Nationale d’Agriculture, Ketou P.O. Box 43, Benin; 2Department of Veterinary Medicine and Animal Production, University of Napoli Federico II, Napoli, F. Delpino 1, 80137 Napoli, Italy; 3Faculty of Agricultural Sciences, University of Abomey-Calavi, Cotonou P.O. Box 2839, Benin

**Keywords:** drought-tolerant plants, *Andropogon virginicus*, *Cenchrus biflorus*, *Zornia latifolia*, in vitro organic matter degradability, drylands, animal feeding, volatile fatty acids, sustainable urban agriculture

## Abstract

**Simple Summary:**

Climate change has increased the frequency of drought along the coastal zone of West Africa, resulting in fodder shortage during the dry season. As forage cropping and conservation are not practiced in this area, animals rely on forage species remaining during the dry season. We assess the vegetation to characterize the range of forage species available for ruminants during the dry season in this region. Among the thirty-three plants consumed by the cows, only thirteen species were available and highly consumed during the dry season. Results showed that most of the forage tested, particularly Poaceae, were of poor in nutritional value; however, the cultivation of some promising drought-tolerant plants such as *Dactyloctenium aegyptium*, *Zornia latifolia*, and *Chamaecrista rotundifolia* could sustain ruminant production along the coastal areas.

**Abstract:**

Along the coast of West Africa, grazing ruminants rely on perennial forage species remaining in uncultivated plots, roadsides, and marshlands during the dry season. To assess the quality of these forages, thirteen drought-tolerant plants were harvested at the mature stage, and the samples were evaluated for chemical composition, in vitro fermentation characteristics, and metabolizable energy (ME) content. They are ten drought-tolerant grasses, including: *Andropogon virginicus*, *Brachiaria deflexa*, *Cenchorus biflorus*, *Dactyloctenium aegyptium*, *Eragrostis tremula*, *Leptochloa caerulescens*, *Loudetia aroundinacea*, *Paspalum notatum*, *Paspalum vaginatum*, *Pennisetum purpureum*, two perennial herbs, *Chamaecrista rotundifolia*, *Zornia latifolia*, and one multipurpose tree, *Elaeis guineensis*. Legume species had the highest nutritional value (highest crude protein and ME, and lowest neutral detergent fiber) of the species studied. In terms of the in vitro data, the gas produced after 120 h of incubation ranged from 149 mL/g in *E. tremula* to 185 mL/g in *Paspalum*. *Z. latifoliaa* and had the fastest rate of fermentation, producing half of the total gas in 19.5 h, whereas *E. tremula* required 49.9 h (*p* < 0.01). The production of branched-chain fatty acids (isobutyrate and isovalerate) was greatest for *E. guineensis* and the lowest in both Paspalum species (*p* < 0.01). The study suggests the need for the protein supplementation of the animals to ensure maximum forage utilization and to satisfy the nutrient requirements of ruminant livestock.

## 1. Introduction

The coastal area is a marginal region for crop production, but there is interest in dairying to provide the growing coastal cities with meat, milk, and milk products, and improve the livelihood of urban farmers [1,2,3]. Manure deriving from cattle improves sandy soil fertility, increasing the yields of food crops, commercial vegetables, and coconut trees. In turn, there is a great opportunity to valorize coastal grasslands under coconut trees and in marshlands by grazing herds, providing the urban market with more dairy products [1].

Many peri-urban dairy herds exist in West African coastal areas, such as Benin, providing about one-third of the milk consumed in and around the cities [3]. A displacement of rural farms around the cities is observed [4,5], increasing the number and size of peri-urban cattle herds to meet the growing demand for milk and milk products. In these farms, as in Sub-Saharan African (SSA) ones, natural pastures are the main feed resource for grazing ruminants, even during the dry seasons [6,7]. In Benin, due to urbanization, this vegetation has transformed into traditional agroforestry systems (fallows, fields, and plantations) and human settlements. The reduction of grazing areas due to urbanization and the ongoing climatic change has affected forage species availability and quality, especially during the dry season [8]. Therefore, the coastal cattle herds, as in West African peri-urban farms [9,10], face severe feeding constraints during the dry seasons, decreasing the animals’ productivity.

Coastal vegetation comprises semideciduous forests on marshlands and sandy soils, offering a diversity of plant species. Mainly, perennial plants available in the coastal sandbanks and wetlands [11] are grazed by ruminant herds. Coastal species described by these authors were composed of weeds and plants resistant to salinity, water, and nutrient stress. Osland et al. [12] depicted that coastal wetland plants are sensitive to climate change, even to small changes in temperature or precipitation. Therefore, in the dry season, only the most resilient plants remained in grasslands for the grazing ruminant herds.

Recently, there has been an increased interest in climate-resilient forage species for improving animal productivity [13]. Most studies have focused on describing some drought-tolerant plants, mainly legumes [14], and their response diversity to abiotic stress [15]. Investigation in the coastal area offers the possibility to identify less-known tropical drought-tolerant forage species, with the most interesting ones regarding their nutritional characteristics. These species could be further investigated for their physiological response to heat stress and cultivation potentialities.

The nutritional problem of livestock during the dry season in SSA [6] is also relevant in Benin coastal areas. Indeed, forage availability in tropical grasslands depends on climatic conditions (i.e., rainfall), while its quality is influenced by the stage of maturity, botanical species, and soil fertility [8,16]. Tropical forages have a particularly high-cell-wall content at the maturity stage, which may negatively affect digestibility [17]. The low-protein and high-lignified-fiber contents in tropical forage could limit the fermentation in the rumen [18]. Moreover, the feeding value of the grazed forage is known to be influenced by growing habits [19], temperatures [20], and seasons [21].

In Benin, dairy herds around Cotonou are experiencing forage scarcity due to urbanization [3,22] and decreasing grasslands [23], which are changed for the residential purpose [24], in contrast with the increase of cattle herds. During the dry season, ruminants rely on grazing perennial grasses and legumes in uncultivated plots, roadsides, and marshlands [8,22]. Except for the study of Koura et al. [8] on the nutritional value of two perennial grasses in the coastal area, little information is available that can be used to evaluate the nutritional status of grazing herds and to design sustainable management strategies in communal grasslands during the dry season. For this reason, assessing the diversity of forage species preferred by cattle in coastal areas is particularly important.

To define the strategies for sustainable ruminant production, it is necessary to optimize the feeding planes by increasing data on the nutritional characteristics of local resources [25]. The knowledge of the farmers on the characteristics of forage species [26] could help identify climate-resilient forage due to their availability and palatability during drought periods. Understanding the nutritional characteristics of these climate-resilient forage species could allow the identification of the best ones to cultivate for sustaining dairy farming along coastal areas and other areas under land-use pressure and climatic changes.

We assume that the studied forage species in coastal grasslands show large variations in their nutritional value that can be exploited for improving herders’ feeding strategies. Therefore, this study assessed the nutritional value and in vitro fermentation characteristics of climate-resilient forage species commonly consumed by peri-urban ruminant herds in the coastal grasslands of Benin.

## 2. Materials and Methods

### 2.1. Study Area

Forage samples were collected in the peri-urban area of Cotonou, which includes the coastal belt of the municipalities of Abomey-Calavi, Ouidah, and Seme-Podji. This area is geographically located between 6.15° and 6.42° north latitude and between 2.00° and 2.15° east latitude. It has a subequatorial climate with two rainy seasons (a long one from April to July and a short rainy season from October to November) alternated with two dry seasons (a short dry season from August to September and a long dry season from December to March). The annual rainfall recorded by the National Direction of Meteorology (DNM) between 1997 and 2016 is between 739.6 mm and 2203.3 mm, with an average of 1305.95 mm. The soil is mainly of sandy, ferrallitic, and hydromorphic types. The native vegetation consists of shrubs, grassland swamps, grassy savannah, marshy meadows, and mangrove forests between lakes on the coastal belt of Benin [27]. Most of this vegetation has been transformed into a mosaic of fallows, farmlands (crops and plantations), and human settlements [24]. This area holds sedentary dairy farms that provide Cotonou, the biggest city in Benin, with milk and milk products [3,8]. The cows are owned by urban dwellers who entrust them to people from the Fulani ethnic group that are culturally cattle herders [28].

### 2.2. Forages Selection and Sampling

Identification of climate-resilient forage was made by investigating farmers’ ecological knowledge of forage species. Local ecological knowledge (LEK) is a cumulative body of knowledge gained via practical interrelationships with ecosystems by local resource users over the years [29], and its use may provide solutions that will lead to sustainable use and management of rangelands. Farmers’ LEK of their resources and use had been acknowledged to provide relevant information on rangeland and its management for the definition of the sustainable management of forage resources [30,31]. To assess farmers’ LEK of climate-resilient forage, three focus group discussions were conducted, one in each of the three studied municipalities, in the village where the availability of cattle herds is elevated. Each group discussion constituted of twelve (12) agropastoral farmers randomly selected among cattle keepers, two (02) responsible from the municipality and agricultural services, and five (05) elder and experienced key farmers. Two experts in grassland management and ruminant production conducted discussions in the local language. The discussions focused on drawing a list of the forage species preferred by ruminant herds and identifying the most interesting climate-resilient forage species using two criteria: their high availability and high consumption by ruminant herds during the dry season. The list of preferred forage species was confirmed through field walk transects with some of the farmers in June 2019 during the rainy season, as described by Ouachinou et al. [32].

For assessing the nutritional value of the climate-resilient forage species mentioned by the herders, forage samples were collected in the grazing areas, including roadsides, under coconut plantations, in marshlands, and in fallows. Samples of forage were collected at their maturity phase in August during the short dry season (temperature: 30 °C, rainfall: 64 mm) preceded by a long rainy season. The method described in Bezabih et al. [25] for forage sampling was used. Six transects of 16 km^2^ were made across communal grazing lands, and forage samples (100 g each) were collected from 60 quadrats of 1.0 m^2^, randomly positioned along the transects. The sample cuts (leaves and stems of herbaceous plants) were taken at the upper part of the plants at 5.0 cm above the ground surface. The sixty (60) subsamples of each forage species per area were subsequently pooled; then, only one (1) sample per forage species was used for the laboratory analysis.

Mixtures of young and mature fresh foliage (leaves and stems < 3 mm diameter) of woody plants were harvested in each of the six transects from three randomly selected trees. Plant samples were harvested from at least three branches in the canopy of each tree. The mixtures from the eighteen (18) different subsamples were pooled, and a sample (5 kg) was kept for laboratory analysis.

### 2.3. Chemical Composition

All the samples were oven-dried at 60 °C for 48 h and ground to pass a 1 mm screen (Brabender Wiley mill, Brabender OHG, Duisburg, Germany). The samples were analyzed for dry matter (DM), crude protein (CP), ether extract (EE), and ash according to the standard procedures (ID number: 2001.12, 978.04, 920.39, and 930.05 for DM, CP, EE, and ash, respectively) as suggested by the AOAC [33]. Neutral detergent fiber (NDF) was determined as described by Van Soest, Robertson, and Lewis [34], and acid-detergent fiber (ADF) and acid-detergent lignin (ADL) were determined as described by Goering and Van Soest [35].

### 2.4. In Vitro Gas Production

The fermentation characteristics and kinetics were studied using the in vitro gas production technique (IVGPT) by incubating all the forage samples at 39 °C under an anaerobic condition with buffered rumen fluid [36]. The substrates to test were weighed (1.0004 ± 0.0003 g) in triplicate side-by-side in 120 mL serum flasks, where 74 mL of anaerobic medium (KCl, NaCl, CaCl_2_ MgSO_4_, NH_4_Cl) was added as a buffer. The anaerobiosis was guaranteed during the trial by flushing CO_2_ during the inoculum preparation and by the hermetical seal of the flasks, and the addition in the medium of reducing agent (cisteina-HCl·H_2_O 0.5 g, 0.5 g Na_2_S·9 H_2_O, 940 mL distilled water). The rumen fluid was collected in a prewarmed thermos from a slaughterhouse authorized according to EU legislation [37] from ovine (*Ovis aries*). To avoid the individual variation, six adult sheep fed a standard diet (NDF 45.5% DM and crude protein 12% DM) were selected as donor animals. The collected material was rapidly transported to the Feed Evaluation Laboratory at the Department of Veterinary Medicine and Animal Production in Napoli (Italy), where it was pooled, flushed with CO_2_, filtered through a cheesecloth, and added in each flask (5 mL). Three flasks containing no substrate were incubated as blanks to correct organic matter (OM) degradability and gas and volatile fatty acids (VFAs) production.

Gas production of fermenting cultures was recorded at 2–24 h intervals during the period of incubation (120 h) using a manual pressure transducer (Cole and Palmer Instrument Co., Illinois, USA) calibrated to atmospheric pressure.

The fermentation was stopped at 120 h, and the fermentation liquor was analyzed for pH with a pH meter (model 3030 Alessandrini Instrument glass electrode, Jenway, Dunmow, UK) to verify the correct trend of the fermentation process. At the end of fermentation, the extent of sample disappearance, expressed as organic matter degradability (dOM, %), was determined by the difference of the incubated OM, and the undegraded filtered (sintered glass crucibles; Schott Duran, Mainz, Germany, porosity # 2) residue burned at 550 °C for 5 h. The cumulative volume of gas produced after 120 h of incubation was related to the incubated OM (OMCV, mL/g) and to the degraded OM (Yield, mL/g).

For VFA determination, samples of fermenting liquors were collected at the end of incubation and centrifuged at 12,000 g for 10 min at 4 °C (Universal 32R centrifuge, Hettich Furn Tech Division DIY, Nussloch, Germany). One milliliter of supernatant was then mixed with 1 mL of oxalic acid (0.06 mol). VFAs were measured by gas chromatography (Thermo Quest 8000top Italia SpA, Rodano, Milan, Italy; fused-silica capillary column, 30 m, 0.25 mm ID, 0.25 µm film thickness), using an external standard solution composed of acetic, propionic, butyric, isobutyric, valeric, and isovaleric acids. The branched-chain fatty acid proportion (BCFA) was calculated as follows: (isobutyric acid+ isovaleric acid)/tVFAs.

### 2.5. Data Processing

The nutritive value of forages was estimated as metabolizable energy (ME, MJ/kg DM) using the equation proposed by Menke and Steingass [38]:$$ME=2.2+0.1357\times GP+0.0057\times CP+0.0002859\times CP^{2}$$
where CP is the content (g/kg DM) of the crude protein and GP is the gas obtained in vitro (mL/200 mg incubated DM) after 24 h of incubation.

For each flask, the gas production profiles were fitted to the sigmoid model described by Groot et al. [39]:$$G=A/[1+{\left(\frac{B}{t}\right)}^{C}]$$
where G is the total gas produced (mL/g of OM) at time t (h), A is the asymptotic gas production (mL/g of OM), B (h) is the time at which one-half of the asymptote is reached, and C is the switching characteristic of the curve. Maximum fermentation rate (Rmax, mL/h) and the time at which it occurs (Tmax, h) were also calculated according to the following formulas [40]:$$Rmax=\left(A\times C^{B}\right)\times B\times \left[Tmax^{\left(-B-1\right)}\right]/\left[\left(1+C^{B}\right)\times {\left(Tmax^{-B}\right)}^{2}\right]$$
$$Tmax=C\times {\left[\frac{B-1}{B+1}\right]}^{\left(\frac{1}{B}\right)}$$

### 2.6. Statistical Analysis

For the statistical analysis, all fermentation characteristics (OMCV, Yield, dOM, pH, VFAs, A, B, Tmax, Rmax) were subjected to an analysis of variance using PROC GLM and SAS [41] according to the model:yij = μ + Fi + εij
where y is the single data, μ is the mean, F is the forage effect (i = 13), and ε is the error term. The minimum significant difference (*p* < 0.01 and *p* < 0.05) was used to verify the differences between means using the Tukey test.

The correlation between the chemical parameters and the in vitro fermentation data was also studied using PROC CORR and SAS [41].

## 3. Results

### 3.1. Diversity of Forage Species Preferred by Dairy Cattle along the Coastal Area of Benin

A diversity of forage species is available and grazed by dairy cows along the coastal area of Benin (Table 1). Thirty-three cattle forage species distributed in eight families were recorded. Poaceae (65%) and Fabaceae (9%) were the most dominant plant families (Figure 1).

The climate-resilient forage species reported by herders as preferred by cattle during the dry season included thirteen plant species: ten Poaceae species (*Andropogon virginicus*, *Brachiaria deflexa*, *Cenchorus biflorus*, *Dactyloctenium aegyptium*, *Eragrostis tremula*, *Leptochloa caerulescens*, *Loudetia aroundinacea*, *Paspalum notatum*, *Paspalum vaginatum*, and *Pennisetum purpureum*), two herbaceous Fabaceae species (*Chamaecrista rotundifolia* and *Zornia latifolia*), and one Arecaceae fodder (*Elaeis guineensis*).

### 3.2. Chemical Composition

The nutritional value of the forage species studied is reported in Table 2. Significant differences were found (*p* < 0.01) in the parameters analyzed. In particular, large variations were found in the CP (average 8.51 ± 3.34% DM) and NDF (average 69.6 ± 11.4% DM). As a consequence of the high CP variation, the estimated ME also varied significantly (average 6.69 ± 2.23 MJ/kg DM). The forage with the most favorable nutritive characteristics was *Z. latifolia* (CP: 14.90% DM; ME: 11.19 MJ/kg DM, NDF: 43.97% DM, EE 5.28% DM; *p* < 0.01), notwithstanding its high content in ADL (10.31% DM; *p* < 0.01). *P. purpureum* showed the lowest metabolizable energy content (4.32 MJ/kg DM; *p* < 0.01).

### 3.3. Fermentation Characteristics

The pH values (mean. 6.78) registered at the end of the incubation fall within the range indicated as adequate for rumen microbial activity. The in vitro fermentation characteristics of the forage species are reported in Table 3. Limited variations were observed for the dOM (average 53.76 ± 5.71%) and OMCV (average 169.2 ± 11.5 mL/g) values. The highest value of the OM degradability was observed in *P. vaginatum*, while the other species showed low degradability, with the lowest value in *E. tremula.* The OMCV and dOM were correlated (*r* = 0.575; *p* < 0.05).

In Figure 2, the in vitro fermentation process during the 120 h of incubation is depicted. Considering that the main differences were observed during the first 24 h of incubation, as shown in Figure 3, these first 24 h of the incubation have been represented. In particular, Fabaceae (*Z. latifolia* and *C. rotundifolia*), followed by Poaceae (*Paspalum* spp.), showed the faster fermentation processes (*p* < 0.01), whereas *P. purpureum* and *E. tremula* had the slowest processes: Rmax 6.25 vs. 2.78 mL/h and Tmax 4.90 vs. 19.0 h (data not shown) for *Z. latifolia* and *P. purpureum*, respectively. The other Poaceae, such as *D. aegyptium*, *C. biflorus*, and *L. caerulescens*, on the one hand, and *A. virginicus*, *B. deflexa*, and *L. aroundinacea*, on the other hand, showed similar trends.

As final fermentation products, the total VFAs, ranging from 59.1 to 87.4 mM/g, was not statistically different (*p* > 0.05) among species, and the branched-chain fatty acids (iso-butyrate and isovalerate) proportion was significantly (*p* < 0.01) highest in *E. guineensis* (0.049 mM/g) and lowest in *P. vaginatum* (0.020 mM/g).

The correlation between the IVGPT parameters and the chemical composition of the forage species (Table 4) confirmed the influence of the chemical composition on the fermentation parameters. The OM degradability was significantly (*p* < 0.001) correlated with ash (*r* = 0.67). The kinetic parameters (B, Tmax, Rmax) were significantly affected by the CP, EE, NDF, ADF, and ADL. The BCFA result significantly correlated to the CP (*r* = 0.50, *p* < 0.01).

## 4. Discussion

### 4.1. Diversity of Forage Species Consumed by Dairy Cows along the Coastal Areas

A large range of forage species is usually consumed in heterogeneous grasslands [31,43] and in the coastal areas. Most of them are grasses surviving in this particular ecosystem under water and nutrient stress and high salinity, which may result in the low productivity of the plant species [12]. The use of true mangrove species, such as *Rhizophora racemosa* and *Avicennia africana* as feed for ruminants [22,44], was not mentioned by the herders; however, their foliage could be used as salt-enrichment nutrients [45].

As the grazed species were widespread weeds, livestock grazing in postharvest cultivated plots could act as biological weed control, as mentioned in previous studies [46,47]. However, the low availability of shrubs and trees in the coastal rangelands limit the utilization of woody plants as a fodder source. Only cultivated *E. guineensis* leaves were used, which may provide the animal with protein and some bioactive compounds, as commonly found in tropical tree leaves [48].

Among the various strategies used to face feed scarcity in Sub-Saharan Africa [49], exploiting available forage species in the dry season is common in the coastal areas. Our study was useful in identifying, together with the herders, some climate-resilient forage species. Indeed, herders of the Fulani ethnic group are well-known to have a good understanding of vegetation, forage species, and their characteristics [32,50].

Most of the climate-resilient forage identified were widespread species in degraded areas (*D. Aegyptium*, *A. virginucus*, *E. tremula*, *L. caerulescens*, *L. arundinacea*, and *P. purpureum*) and species from wet and saline habitats (e.g., *Paspalum* spp.). However, some plants were confirmed to be drought-tolerant (*C. rotundifolia* and *Z. latifolia*) and grasses (*B. deflexa* and *C. biflorus*) were elicited by the herders [42]. Indeed, *B. deflexa* and *C. biflorus* have been used as food plants during severe famines [51]. These perennial grass and herb plants could be potential drought-tolerant plants worthy of further investigation for their value in the coastal areas and other areas facing severe climatic stress such as the arid and semiarid zones of West Africa. Moreover, perennial legumes could help to improve soil fertility.

### 4.2. Chemical Composition of Forage Consumed by Dairy Cattle

Previous studies have investigated some of the studied species cultivated in Benin [52] or other tropical conditions [25,26,53,54,55]. Moreover, some data are found in the literature on the chemical composition and in vitro organic matter degradability of *B. deflexa*, *D. aegyptium*, and *E. guineensis* in the sub-humid area [56]. However, our study was important to evaluate the OM degradability of the tropical grasses and legumes at their maturity stage; according to Calabrò et al. [53], harvest forages at the late stage may affect the nutritive value and animals’ performance. This information could help enhance the sustainable utilization of these forage species for improving ruminant performance on coastal grasslands during the dry season.

Comparing the crude protein content of a previous study [52] on the cultivated forage of the same area, a lower protein content in the natural forage was observed. Indeed, Musco et al. [52] did not state the stage of maturity of the studied forages; their forages may have been less mature, resulting in higher protein levels. The difference in the sampling area, forage type, genotype, soil, and management conditions are also known to influence the nutritional value of forage species [8,16,57]. However, the nutritional contents found in our forage are similar to the findings of Michiels et al. [58] in other subhumid areas in dry seasons.

The high structural carbohydrates (NDF and ADF) and low crude protein (4.92 to 8.42% DM) found in Poaceae species is comparable to data on cultivated forage in semiarid areas [26], and the maturity stage at harvest of our samples can explain these results [59]. The poor soil conditions [60] in these coastal areas of Benin, where sandy soils of low-fertility levels are dominant [61], could also explain the low-protein content in the studied forage.

Regarding grass species, the particularly higher nutritive value of *D. aegyptium* compared to other grasses was also reported [8,26], revealing this widespread grass as a promising one among the poor grasses available in the coastal areas. On the other hand, the low-protein and energy content in *E. tremula* was previously reported [53] in semiarid areas of Niger.

### 4.3. In Vitro Fermentation Characteristics

Large variation in the chemical composition of the forage samples was reflected in fermentation parameters, as found in tropical forage [16]. Several significant correlations between chemical and fermentation parameters were observed, as the nutritional characteristics affect the availability of nourishing substances for ruminal bacteria [52]. Among the two *Paspalum* species, high OM degradability was obtained in *P. vaginatum*, confirming data in the literature [55], while *P. notatum* had shown the highest nutritional value. The relatively high nutritional value of *P. notatum* compared to other grasses is acknowledged [62]; however, the species is not very productive in natural grasslands. Despite the high-ether-extract and lignin content, *Z. latifolia* and *E. guineensis* showed high values of OM degradability, likely as a result of the high-crude-protein content and low-NDF content. Considering the high content of antinutritional factors common in tropical legumes [52], the ADL content may have probably been overestimated, as Marles [63] reported.

On the other hand, it is more difficult to justify why the high-ether-extract content did not negatively affect the dOM and OMCV, since lipids are indicated as limiting factors in rumen fermentation [64]. However, among Leguminosea, *E. guineensis* showed a slow in vitro fermentation rate despite its favorable chemical composition (crude protein, NDF, and energy content), probably due to some antinutritional factors; the latter was depicted by Ibrahim and Jaafar [65]. Overall, compared to spontaneous forage species in semiarid areas of Niger [53], the fermentation rate of Leguminosae and of the forage tree is greater in our study.

The positive correlation between the CP and fatty acids (isovalerate and valerate), as well as the negative correlation of NDF with the OM degradability, has been observed [66]. The potential negative correlation of gas production (Yield) with crude protein (−0.37, *p* < 0.05) and ash (−0.63, *p* < 0.001) was found; these nutrients could negatively interfere with microbial activity, as reported by Getachew et al. [66] and Musco et al. [52].

As generally observed in IVGPT studies, and also in this trial, a significant correlation between fermentation parameters was observed (data not shown): a highly significant (*p* < 0.01) correlation coefficient was found between VFAs with dOM and OMCV. These data also indicate the validity of the in vitro study for these kinds of feeds and confirm that the recorded gas gives useful information on the availability of energy provided by the feeds and made available to animals.

### 4.4. Implications for Sustainable Dairy Cows’ Production in Coastal Areas

The poor nutritional value, low crude protein, and high NDF in Poaceae suggested the need for protein supplementation to meet the nutrient requirements of ruminants, in particular during the dry season. Fodder trees are known to be used in various tropical environments for protein supply to animals [57,67], but they are less available. The intensification process ongoing in urban dairy farms in the Sahelian areas [68] with stall-fed animals supplemented by using agro-processing byproducts and concentrates would be an issue in peri-urban coastal dairy farms. However, using byproducts may result in higher input than the traditional farming methods currently used in coastal dairy farms. Although, there could be a great opportunity to improve forage availability through cropping [69] by using climate-resilient forage species. So far, research on forage species to cultivate in SSA had focused on dual-purpose cereals and legumes [70]; however, our study showed some interesting native grass and legume species to be introduced in agricultural systems.

Indeed, increasing the availability of the best forage (high CP and ME, low NDF, high fermentation rate), such as Fabaceae (*Z. latifolia*, *C. rotundifolia*) and Poaceae (*D. aegyptium*, *P. notatum*, *C. biflorus*), seems a good strategy in the coastal area. These species could be cultivated in alley-cropping with commercial vegetables or under coconut plantations [71], as suggested by the coastal herders in recent investigations [1]. These plants could be grown at the end of the rainy season, allowing utilization at the early vegetation stage of forage of high quality in the dry season. In particular, *C. biflorus* needs to be grazed at the earlier vegetative stage, as the mature plant is pungent. *D. aegyptium*, as a low-methanogenesis plant [8], could be the most promising Poaceae for sustaining ruminant production along the coastal area.

Forage cropping in urbanized areas would be limited by the land shortage, lack of knowledge, and need for additional labor [72]. Training the farmers to improve their awareness and capacities would improve adoption rates. Other ecosystem services of forage cultivation, such as decreasing erosion and improving soil fertility [73], could be emphasized. For the ones with low land availability, legume (*Z. latifolia* and *C. rotundifolia*) cultivation in small pieces of land to improve protein supply could be the best choice. This also opened the way for the development of fodder markets in urban areas, as Konlan et al. [9] suggested, where crop residues, agro-processing by-products, and fresh forage could be sold.

## 5. Conclusions

Exploring the nutritive value and in vitro fermentation characteristics of climate-resilient forage species collected in natural grasslands showed the low nutritive characteristics. Among Poaceae, *Dactyloctenium aegyptium*, *Paspalum notatum*, and *Cenchrus biflorus* showed the best nutritional characteristics. *Zornia latifolia* was the best among the legumes. Our study allows us to assess, for the first time, the nutritional potential of less-known species such as *Andropogon virginicus*, *Cenchrus biflorus*, and *Z. latifolia*. The cattle farmers are encouraged to undertake conservation management and the deliberate production of preferred forage species such as *D. aegyptium*, *Z. latifolia*, and *Chamaecrista rotundifolia*. By doing so, forage availability would increase and nutritional value would be improved for sustaining livestock feeding in the periurban dairy farms of West Africa. Further studies should focus on the other nutritional characteristics of the studied forages (i.e., antinutritional factors), including the in vivo evaluation of ruminants.

## Figures and Tables

**Figure 1 animals-12-03550-f001:**
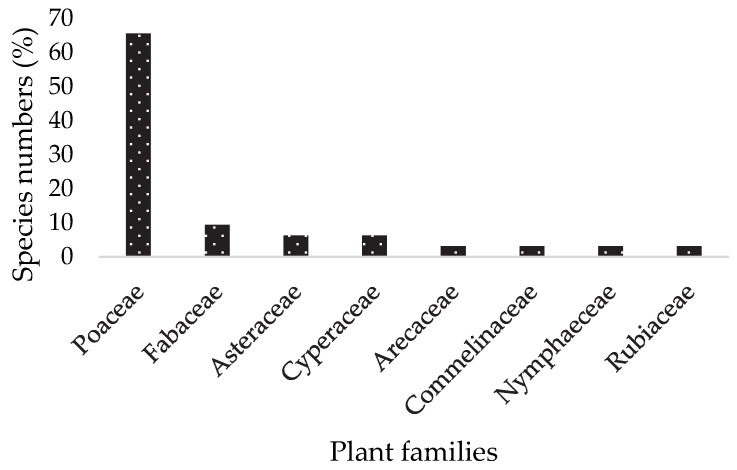
Plant families in the study sites.

**Figure 2 animals-12-03550-f002:**
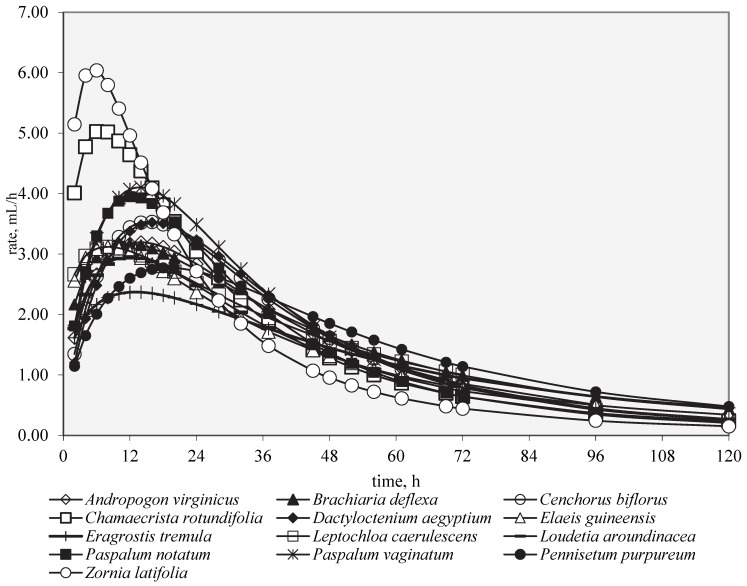
In vitro fermentation rate over time (120 h) for forage species.

**Figure 3 animals-12-03550-f003:**
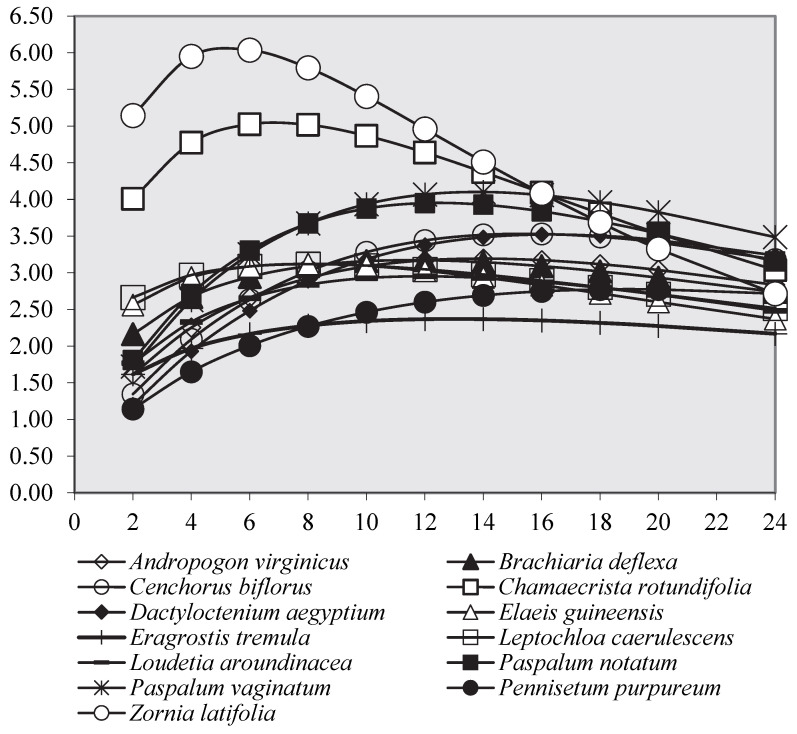
In vitro fermentation rate over time (24 h) for forage species.

**Table 1 animals-12-03550-t001:** Forage species consumed by cattle during the dry season along the coastal region of Benin.

Species	Local Name	Family	Life Form *	Common Name	Climate Resilience **
*Acroceras zizanioides*	Pagouri (Fu)	Poaceae	Perennial grass	Oat grass	no
*Andropogon virginicus*	Klogbou (Fo)	Poaceae	Perennial grass	Broomsedge	yes
*Brachiaria deflexa*	Rôti (Yo)	Poaceae	Annual grass	Guinea millet	yes
*Cenchrus biflorus*	Agbokodjagbé (Fo)	Poaceae	Annual grass	Cram-cram	yes
*Centrosema puberscens*	Gadigui (Fu)	Fabaceae	Perennial herb	Butterfly pea	no
*Chamaecrista rotundifolia*	Abèko (Yo)	Fabaceae	Perennial or annual plant	Round-leaf cassia	yes
*Commelina benghalensis*	Balassa (Fu)	Commelinaceae	Annual or perennial herb	Wandering jew	no
*Cyperus difformis*	Kponnikpon (Fo)	Cyperaceae	Annual grass	Small-flowered nutsedge	no
*Dactyloctenium aegyptium*	Landalaho (Fu)	Poaceae	Perennial grass	Crowfoot grass	yes
*Echinochloa colona*	Goal (Fo)	Poaceae	Annual grass	Junglerice	no
*Elaeis guineensis*	Déman (Fo)	Arecaceae	Woody	African oil palm	yes
*Elytrophorus spicatus*	Gan siri (Fu)	Poaceae	Annual or perennial plant	Spikegrass	no
*Eragrostis pilosa*	Selsênin (Fu)	Poaceae	Annual grass	India lovegrass	no
*Eragrostis tremula*	Sarao	Poaceae	Annual grass or perennial	Chinese lovegrass	yes
*Ischaemum rugosum*	Hêbêrê	Poaceae	Annual grass	Saramolla grass	no
*Kyllinga erecta*	Goal	Poaceae	Annual grass	spikesedges	no
*Kyllinga squamulata*	Goal	Poaceae	Annual grass	Crested greenhead sedge	no
*Leptochloa caerulescens*	Monlougbé (Fo)	Poaceae	Annual grass	Sprangletops	yes
*Loudetia arundinacea*	Kékéyo (Yo)	Poaceae	Perennial grass	Russet grass	yes
*Mariscus cylindristachyus*	Gbékui (Fo)	Cyperaceae	Annual grass	Flatsedges	no
*Nymphaea lotus*	Flowa (Fu)	Nymphaeceae	Perennial herb	White Egyptian lotus	no
*Oryza barthii*	Rayêrê (Fu)	Poaceae	Annual grass	African wild rice	no
*Panicum maximum*	Gayéri (Fu)	Poaceae	Annual grass	Guinea grass	no
*Paspalum notatum*	Gazongbé (Fo)	Poaceae	Perennial grass	Bahiagrass	yes
*Paspalum vaginatum*	Tchitchiri (Fu)	Poaceae	Perennial grass	Seashore paspalum	yes
*Pennisetum pedicellatum*	Hulunin (Fu)	Poaceae	Perennial grass	Deenanath grass	no
*Pennisetum purpureum*	Fan vovo (Fo)	Poaceae	Perennial grass	Napier grass	yes
*Spermacoce verticillata*	Goudoudél (Fu)	Rubiaceae	Annual or perennial	Shrubby false bottoweed	no
*Synedrella nodiflora*	Badjanadji (Fu)	Asteraceae	Perennial herb	Nodeweed	no
*Tridax procumbens*	Kourkoudi (Fu)	Asteraceae	Perennial herb	Coat buttons	no
*Vossia cuspidata*	Talol (Fu)	Poaceae	Aquatic grass	Hippo grass	no
*Zornia latifolia*	Linguéri (Fu)	Fabaceae	Perennial herb	Maconha brava	yes

Local languages: Fu.: Fulani, Fo.: Fongbé, Yo.: Yoruba. * Akobundu and Agyakwac [11]; CABI [42]. ** Climate-resilient, as perceived by herders.

**Table 2 animals-12-03550-t002:** Nutritional value of the forage species (*n* = 26).

Forage Species	DM	Ash	CP	EE	NDF	ADF	ADL	ME ^1^
		% DM						MJ/kg DM
Poaceae								
*Andropogon virginicus*	90.82 ^ab^	11.57 ^ab^	7.84 ^cd^	1.10 ^ef^	73.88 ^cd^	47.52 ^ab^	5.85 ^cd^	5.84 ^de^
*Brachiaria deflexa*	91.03 ^ab^	8.1 ^cde^	7.07 ^de^	1.31 ^def^	75.92 ^bc^	41.87 ^cd^	-	5.56 ^ef^
*Cenchorus biflorus*	90.35 ^ab^	8.22 ^cde^	7.83 ^cd^	1.39 ^def^	73.81 ^cd^	46.18 ^bc^	7.06 ^bc^	5.93 ^de^
*Dactyloctenium aegyptium*	88.45 ^b^	10.03 ^bcd^	8.23 ^cd^	1.79 ^c^	70.61 ^e^	46.33 ^bc^	7.60 ^b^	5.98 ^cd^
*Eragrostis tremula*	91.75 ^a^	4.99 ^f^	6.45 ^ef^	1.16 ^def^	81.80 ^a^	-	-	5.13 ^g^
*Leptochloa caerulescens*	91.33 ^a^	8.91 ^bcde^	5.51 ^fg^	1.27 ^def^	78.88 ^ab^	46.77 ^b^	4.66 ^d^	5.33 ^fg^
*Loudetia aroundinacea*	91.06 ^ab^	10.84 ^abc^	6.53 ^ef^	1.04 ^f^	73.88 ^cd^	51.24 ^a^	4.00 ^d^	5.25 ^fg^
*Paspalum notatum*	91.36 ^a^	6.83 ^def^	8.42 ^c^	1.43 ^cde^	71.19 ^de^	41.95 ^c^	7.64 ^b^	6.62 ^c^
*Paspalum vaginatum*	90.26 ^ab^	13.12 ^a^	6.09 ^ef^	1.40 ^def^	73.93 ^cd^	47.55 ^ab^	7.07 ^bc^	5.60 ^ef^
*Pennisetum purpureum*	91.11 ^a^	7.79 ^def^	4.62 ^g^	1.47 ^cd^	78.45 ^b^	51.27 ^a^	5.61 ^cd^	4.32 ^h^
Fabaceae								
*Chamaecrista rotundifolia*	90.99 ^ab^	5.99 ^ef^	13.19 ^b^	3.34 ^b^	54.79 ^f^	-	-	9.84 ^b^
*Zornia latifolia*	89.56 ^ab^	9.41 ^bcd^	14.90 ^a^	5.28 ^a^	43.97 ^h^	38.31 ^de^	10.31 ^a^	11.19 ^a^
Arecaceae								
*Elaeis guineensis*	90.32 ^ab^	9.84 ^bcd^	14.01 ^a^	1.10 ^ef^	53.66 ^g^	36.97 ^e^	8.35 ^b^	10.36 ^b^
MSE	5.73	6.26	0.25	0.08	0.83	0.97	0.33	0.15

-: less common forage species; not enough material was available to complete all the analysis. MSE: mean square error. Along the column, for each fodder, lower-case letters indicate statistically significant differences (*p* < 0.01). DM: dry matter; CP: crude protein; EE: ether extract; NDF: neutral-detergent fiber; ADF: acid-detergent fiber; ADL: acid-detergent lignin; ME: metabolizable energy. ^1^ Metabolizable energy was estimated using the equation proposed by Menke and Steingass [38]; *n* = 26 corresponded to 13 forages × 02 replicates.

**Table 3 animals-12-03550-t003:** In vitro fermentation characteristics and volatile fatty acids production of the forage species (*n* = 39).

Forage Species	pH	dOM	OMCV	Yield	A	B	tVFAs	BCFA
		%	mL/g	mL/g	mL/g	h	mM/g iOM	
Poaceae								
*Andropogon virginicus*	6.83	51.03 ^cd^	166.35 ^bc^	325.98 ^abcd^	200.74 ^bcde^	38.25 ^cde^	72.52	0.025 ^bcd^
*Brachiaria deflexa*	6.70	55.27 ^bc^	175.38 ^ab^	317.74 ^bcd^	223.88 ^abc^	44.14 ^bc^	77.65	0.025 ^bcd^
*Cenchorus biflorus*	6.82	51.90 ^cd^	172.96 ^ab^	333.41 ^abc^	198.39 ^cde^	34.49 ^defg^	78.36	0.026 ^bcd^
*Dactyloctenium aegyptium*	6.86	55.09 ^bc^	174.57 ^ab^	317.07 ^bcd^	198.66 ^bcde^	34.83 ^def^	76.19	0.027 ^bcd^
*Eragrostis tremula*	6.80	42.30 ^e^	148.65 ^d^	351.61 ^ab^	210.58 ^bcd^	55.69 ^a^	62.70	0.034 ^b^
*Leptochloa caerulescens*	6.81	54.28 ^bc^	175.73 ^ab^	323.77 ^abcd^	245.31 ^a^	52.16 ^ab^	74.80	0.023 ^cd^
*Loudetia aroundinacea*	6.60	51.38 ^cd^	150.11 ^cd^	292.21 ^def^	180.51 ^e^	37.33 ^cde^	59.13	0.035 ^b^
*Paspalum notatum*	6.85	52.35 ^cd^	172.68 ^ab^	329.69 ^abc^	190.93 ^de^	27.46 ^fgh^	79.17	0.027 ^bcd^
*Paspalum vaginatum*	6.68	66.87 ^a^	184.69 ^a^	276.05 ^ef^	210.78 ^bcd^	31.50 ^efg^	87.37	0.020 ^d^
*Pennisetum purpureum*	6.76	48.92 ^d^	173.60 ^ab^	355.05 ^a^	227.67 ^ab^	49.94 ^ab^	70.12	0.026 ^bcd^
Fabaceae								
*Chamaecrista rotundifolia*	6.82	52.94 ^cd^	171.66 ^ab^	324.27 ^abcd^	211.97 ^bcd^	26.12 ^gh^	77.52	0.032 ^bc^
*Zornia latifolia*	6.90	58.45 ^b^	180.38 ^ab^	309.18 ^cde^	194.90 ^cde^	19.48 ^h^	76.38	0.029 ^bcd^
Arecaceae								
*Elaeis guineensis*	6.71	58.16 ^b^	152.52 ^cd^	262.49 ^f^	199.12 ^bcde^	40.90 ^cd^	69.29	0.049 ^a^
MSE	0.10	1.49	5.70	12.10	8.03	2.35	20.91	0.003

MSE: mean square error. Along the column, for each fodder, lower-case letters indicate statistically significant differences (*p* < 0.01). dOM = organic matter degradability (% of incubated OM); OMCV = cumulative volume of gas related to incubated OM; Yield = cumulative volume of gas related to degraded OM; A = potential gas production; B = time at which A/2 was formed; BCFA: branched-chain fatty acid proportion on tVFAs; tVFAs: total volatile fatty acids. *n* = 39 corresponded to 13 forages × 03 replicates.

**Table 4 animals-12-03550-t004:** Significance of correlation between some chemical parameters and in vitro fermentation data (*n* = 39).

	dOM	OMCV	Yield	B	Tmax	Rmax	Isobut	Isoval	Val	BCFA
	%	mL/g	mL/g	h	h	mL/h	mM/g iOM			
Ash	0.67 ***	0.21 NS	−0.63 ***	−0.22 NS	0.06 NS	0.01 NS	−0.04 NS	−0.12 NS	−0.07 NS	−0.18 NS
CP	0.28 NS	−0.03 NS	−0.37 *	−0.62 ***	−0.60 ***	0.66 ***	0.40 *	0.42 *	0.59 ***	0.50 **
EE	0.23 NS	0.37 *	0.01 NS	−0.65 ***	−0.48 **	0.8 ***	0.09 NS	0.07 NS	0.38 *	−0.04 NS
NDF	−0.40 *	−0.10 NS	0.42 *	0.69 ***	0.59 ***	−0.74 ***	−0.36 *	−0.36 *	−0.57 ***	−0.41 *
ADF	−0.36 NS	−0.02 NS	0.35 NS	0.40 *	0.59 ***	−0.50 **	−0.38 *	−0.38 *	−0.57 ***	−0.44 *
ADL	0.45 *	0.31 NS	−0.21 NS	−0.71 ***	−0.33 NS	0.74 ***	0.28 NS	0.27 NS	0.50 **	0.19 NS

CP = crude protein, EE = ether extract, NDF = neutral-detergent fiber, ADF = acid-detergent fiber, ADL = acid-detergent lignin; dOM = organic matter degradability, OMCV = cumulative volume of gas related to incubated OM, Yield = cumulative volume of gas related to degraded OM, B = time at which A/2 was formed, Tmax = time at which maximum rate was reached, Rmax = maximum fermentation rate, isobut = isobutyrate, isoval = isovalerate, val = valerate, BCFA = branched-chain fatty acids proportion. *, **, ***, NS: *p* < 0.05, *p* < 0.01, *p* < 0.001, and not significant, respectively.

## Data Availability

The datasets used and analyzed during the current study are available from the corresponding author upon reasonable request.

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
