# Peer review of "Nutritional Value of Climate-Resilient Forage Species Sustaining Peri-Urban Dairy Cow Production in the Coastal Grasslands of Benin (West Africa)"

_animals, 2022, doi:10.3390/ani12243550_

Round 1

Reviewer 1 Report

The subject dealt with is certainly important for the nutrition of animals reared in extensive systems: evaluating the NV of palatable fodder in the dry season when there is a loss of nutrients. I would have liked to see the analysis of samples at different intervals during the season to see the variation, something the authors could see in perspective.

I noted some corrections with respect to the writing of the scientific names of the species after the first appearance; the short writing must be initial of the name of the genus and in full the name of the species: the author sometimes wrote the name of the genus alone. Example: Eragrostis tremula, write in short E. tremula

Author Response

Responses to Reviewer 1 comments

We would like to first thank the reviewer for his valuable comments. We have responded point-by-point to all the comments and hope to have provided details accordingly.

Comment. The subject dealt with is certainly important for the nutrition of animals reared in extensive systems: evaluating the NV of palatable fodder in the dry season when there is a loss of nutrients. I would have liked to see the analysis of samples at different intervals during the season to see the variation, something the authors could see in perspective.

Author response. We thank the reviewers for these remarks. We will consider the suggested perspective in a future study.

Comment. I noted some corrections with respect to the writing of the scientific names of the species after the first appearance; the short writing must be initial of the name of the genus and in full the name of the species: the author sometimes wrote the name of the genus alone. Example: Eragrostis tremula, write in short E. tremula

Author response. The writing of the scientific names was corrected in the whole manuscript.

Other comments in the attached file

Comment. Line 128. “animal”

Author response. the sentence was revised.

Comment. Line 139. “in one village”

Author response. the sentence was corrected.

Comment. Line 140. “constituted”

Author response. the sentence was corrected.

Comment. Line 158. “60” and “after”

Author response. The sentence was revised.

Comment. Line 163. “eighteen”

Author response. The sentence was revised.

The writing of the scientific names was corrected in the whole manuscript.

Reviewer 2 Report

Perhaps, in future research, the incubation time will be extended to 240 hours. Researches have shown that this time is more adequate to evaluate the indigestible fraction of tropical forages. Additionally, I would recommend correcting the neutral detergent fiber for ash and protein to get a truer representation of the fiber.

Author Response

Responses to Reviewer 2 comments

We would like to first thank the reviewer for his valuable comments. We have responded point-by-point to all the comments and hope to have provided details accordingly.

Comment. Perhaps, in future research, the incubation time will be extended to 240 hours. Researches have shown that this time is more adequate to evaluate the indigestible fraction of tropical forages.

Author response. We thank the reviewer for this suggestion. We will consider this advice in our future studies.

Comment. Additionally, I would recommend correcting the neutral detergent fiber for ash and protein to get a truer representation of the fiber.

Author response. The neutral detergent residues were corrected only for ash. In previous investigation, we verified that Nitrogen linked to ADF is really a very small amount in this kind of forages.

Reviewer 3 Report

Line  Comment

24 at the mature stage,

29  Legume species had the highest nutritional value (highest crude protein and ME and lowest neutral detergent fiber) of the species studied.

32 Zornia had the fastest rate of fermentation, producing half of the total gas in 19.5 hours whereas eragrostis required 49.9 hours (P<0.01).

34  the production of branched-chain fatty acids (isobutyrate and iso-valerate) was greatest for Elaeis and lowest in both paspalum species (P< 0.01)

43 The coastal area is a marginal region for crop production, but there is interest in dairying to provide the growing coastal cities with meat, milk and milk products and improve the livelihood of urban farmers [1, 2, 3}.

81 soil fertility

129  ethnic group

176 weighed

177 were triplicates side by side or completely randomized from each other?

180 2.02 extra?

271 Is this ME derived from gas production? Or calculated from chemical analysis?  Needs a footnote to clarify.

283 Eragrostis.  OMCV and dOM were correlated (r=0.575; p<.05).

314 Would be better to say significantly correlated to CP,…

350  plants are worthy of further investigation for their value in the coastal area…

363 reference 52 did not state the stage of maturity in the paper.  Their forages may have been less mature, resulting in higher protein levels.

371 adbancd maturity stage at harvest of our samples…

390 degradability, likely a result of high crude protein content and low NDF content.

392 ADL content may…

400 tree were greater in our study.

403 negative correlation of NDF with OM digestibility has been observed [66}

404 according to table, CP wa positive influence on dOM.

420 It seems to me that stall fed animals consuming byproducts may be higher input than the traditional farming methods.

451 Our study allows us to assess for the first

Author Response

Responses to Reviewer 3 comments

We would like to first thank the reviewer for his valuable comments. We have responded point-by-point to all the comments and hope to have provided details accordingly.

Comment. 24 at the mature stage,

Author response. The sentence was revised accordingly.

Comment. 29  Legume species had the highest nutritional value (highest crude protein and ME and lowest neutral detergent fiber) of the species studied. Lines 29-32.

Author response. The sentence was revised accordingly.

Comment. 32 Zornia had the fastest rate of fermentation, producing half of the total gas in 19.5 hours whereas eragrostis required 49.9 hours (P<0.01). lines 34-35.

Author response. The sentence was revised accordingly.

Comment. 34  the production of branched-chain fatty acids (isobutyrate and iso-valerate) was greatest for Elaeis and lowest in both paspalum species (P< 0.01). lines 37-38.

Author response. The sentence was revised accordingly.

Comment. 43 The coastal area is a marginal region for crop production, but there is interest in dairying to provide the growing coastal cities with meat, milk and milk products and improve the livelihood of urban farmers [1, 2, 3}.

Author response. The sentence was revised accordingly.

Comment. 81 soil fertility

Author response. The sentence was revised accordingly.

Comment. 129  ethnic group

Author response. The sentence was revised accordingly.

Comment. 176 weighed

Author response. The sentence was revised accordingly.

Comment. 177 were triplicates side by side or completely randomized from each other?

Author response. The three bottles from each forage sample were arranged side by side.

Comment. 180 2.02 extra?

Author response. Details was provided to the formula for reducing agent, as followed: cisteina-HCl·H2O 0.5 g, 0.5 g Na2S·9 H2O, 940 ml distilled water.

Comment. 271 Is this ME derived from gas production? Or calculated from chemical analysis?  Needs a footnote to clarify.

Author response. The ME derived from gas production (Gas produced at 24h) and chemical composition (Crude protein content) as stated in line 223 in the methodology. A footnote was added (line 287).

Comment. 283 Eragrostis.  OMCV and dOM were correlated (r=0.575; p<.05).

Author response. The sentence was revised accordingly.

Comment. 314 Would be better to say significantly correlated to CP …

Author response. The sentence was revised accordingly.

Comment. 350  plants are worthy of further investigation for their value in the coastal area…

Author response. The sentence was revised accordingly.

Comment. 363 reference 52 did not state the stage of maturity in the paper. Their forages may have been less mature, resulting in higher protein levels.

Author response. The sentence was revised accordingly.

Comment. 371 adbancd maturity stage at harvest of our samples…

Author response. The sentence was revised accordingly.

Comment. 390 degradability, likely a result of high crude protein content and low NDF content.

Author response. The sentence was revised accordingly.

Comment. 392 ADL content may…

Author response. The sentence was revised accordingly.

Comment. 400 tree were greater in our study.

Author response. The sentence was revised.

Comment. 403 negative correlation of NDF with OM digestibility has been observed [66}

Author response. The sentence was revised.

Comment. 404 according to table, CP wa positive influence on dOM.

Author response. Yes, the reviewer is right for CP correlation with dOM. The sentence was revised (lines 419-422).

Comment. 420 It seems to me that stall fed animals consuming byproducts may be higher input than the traditional farming methods.

Author response. Yes that is right. The sentence was revised (lines 439-340).

Comment. 451 Our study allows us to assess for the first

Author response. The sentence was revised accordingly.

Reviewer 4 Report

This is a good and interesting paper aimed to assess the nutritional characteristics of many climate-resilient forage species sustaining peri urban dairy cow production in coastal grasslands of Benin (West-Africa).

The nutritional value of forage species was evaluated through the conventional chemical composition analyses as well as in vitro gas production technique to assess forages fermentation characteristics and kinetics.

Novel findings have been reported by Authors and the subject is adequate with the overall journal’s scope.

I would like to congratulate Authors for the good-quality of the article, the literature reported used to write the paper, and for the clear and appropriate structure.

The manuscript is well written, presented and discussed, and understandable to a specialist readership.

The work shows a conscientious study in which a very exhaustive discussion of the literature available has been carried out.

The results have been clearly presented and analyzed exhaustively.

Moreover, the obtained results have been well discussed by using appropriate and recent literature.

As specific comments, I suggest to reduce the length of the Introduction section and to further improve the Conclusion section.

Check also if all references have been cited into the text or reported in the references list.

Few grammar typos need to be corrected.

So, I recommend the acceptance of the manuscript after minor revision.

Author Response

Responses to reviewer 4 comments

We would like to first thank the reviewer for his valuable comments. We have responded point-by-point to all the comments and we hope to have provided details accordingly.

Comment. This is a good and interesting paper aimed to assess the nutritional characteristics of many climate-resilient forage species sustaining peri urban dairy cow production in coastal grasslands of Benin (West-Africa).

The nutritional value of forage species was evaluated through the conventional chemical composition analyses as well as in vitro gas production technique to assess forages fermentation characteristics and kinetics.

Novel findings have been reported by Authors and the subject is adequate with the overall journal’s scope.

I would like to congratulate Authors for the good-quality of the article, the literature reported used to write the paper, and for the clear and appropriate structure.

The manuscript is well written, presented and discussed, and understandable to a specialist readership.

The work shows a conscientious study in which a very exhaustive discussion of the literature available has been carried out.

The results have been clearly presented and analyzed exhaustively.

Moreover, the obtained results have been well discussed by using appropriate and recent literature.

Author response. We thank very much the reviewers for these good words, we appreciate very much. We thank also the reviewer for all his comments that help to improve our manuscript.

Comment. As specific comments, I suggest to reduce the length of the Introduction section and to further improve the Conclusion section.

Author response. The introduction was shorter, and the conclusion improved.

Comment. Check also if all references have been cited into the text or reported in the references list.

Author response. We have checked again. All the references cited are reported in the references list.

Comment. Few grammar typos need to be corrected.

Author response. The whole manuscript was corrected for grammar errors.

Comment. So, I recommend the acceptance of the manuscript after minor revision.

Author response. Thank you very much for your appreciation.
